# The Danish Trauma Database for Refugees (DTD): A Multicenter Database Collaboration—Overcoming the Challenges and Enhancing Mental Health Treatment and Research for Refugees

**DOI:** 10.3390/ijerph20166611

**Published:** 2023-08-20

**Authors:** Marie Høgh Thøgersen, Line Bager, Sofie Grimshave Bangsgaard, Sabina Palic, Mikkel Auning-Hansen, Stine Bjerrum Møller, Kirstine Bruun Larsen, Louise Tækker, Bo Søndergaard Jensen, Søren Bothe, Linda Nordin

**Affiliations:** 1The Danish Institute Against Torture (DIGNITY), 2100 Copenhagen, Denmark; lbager.ncrr@econ.au.dk (L.B.); sgba@dignity.dk (S.G.B.); sapk@dignity.dk (S.P.); sobo@dignity.dk (S.B.); ln@dignity.dk (L.N.); 2National Center for Register-Based Research, Aarhus University, 8210 Aarhus, Denmark; 3The Rehabilitation Center for Trauma Survivors (RCT), 6100 Haderslev, Denmark; mikkel@rct.life; 4The Clinics for Trauma and Torture Survivors (ATT), 7100 Vejle, Denmark; stinebm@health.sdu.dk; 5Department of Psychology, University of Southern Denmark, 5230 Odense, Denmark; 6The Rehabilitation Center for Refugees (RCF), 9000 Ålborg, Denmark; kibrl@rn.dk; 7Privat Treatment Center for Traumatized Refugees and Their Families, (OASIS), 1164 Copenhagen, Denmark; louise@oasis-rehab.dk; 8The PTSD and Anxiety Clinic, Aarhus University Hospital, 8200 Aarhus, Denmark; bojensen@rm.dk; 9Department of Psychology, Lund University, 22100 Lund, Sweden

**Keywords:** database, DTD, multicenter research, refugees, trauma, PTSD, treatment, migration

## Abstract

Mental health of trauma-affected refugees is an understudied area, resulting in inadequate and poorer treatment outcomes. To address this, more high-quality treatment studies that include predictive analyses, long-term evaluations, cultural adaptations, and take account for comorbidities, are needed. Moreover, given the complex intertwining of refugees’ health with post-migration stressors and other social factors, it is crucial to examine the social determinants of refugee mental health. The Danish Trauma Database for Refugees (DTD) is a multicenter research database uniting six national centers that provide outpatient treatment for trauma-affected refugees. Through the database, we collect clinical and sociodemographic data from approximately 1200 refugees annually and will merge the database with Danish population register data. The purpose of the DTD is two-fold; clinical and research. The DTD offers data-driven guidance for routine clinical treatment planning of the individual patient, as well as exceptional research opportunities for testing treatment interventions in clinical settings, with larger sample sizes, and more representative heterogeneity of the population. Complex analyses of risk and protective factors, barriers, access to treatment, and societal and transgenerational aspects of trauma are possible with the DTD. This conceptual paper introduces the DTD, the historical background, the development process and implementation strategy, and the associated challenges with developing and running a multicenter database. Most importantly, it highlights the clinical and research potential of the DTD for advancing the understanding and treatment of trauma-affected refugees.

## 1. Introduction

Refugees often suffer from severe psychiatric and somatic health problems, including posttraumatic stress disorder (PTSD), depression, anxiety, and chronic pain [1,2]. Their poor mental and physical health is often related to traumatic pre-migration events (e.g., war, torture, and other violence), and at the same time, to the severe stress of migration and difficulties associated with settling down in a new country [3,4,5]. Meta-analyses and systematic reviews reveal that refugees suffer from mental disorders up to ten times more often than the general population [6]. Despite this, access to mental health services remains inadequate due to barriers such as culture, language, help-seeking behaviors, and lack of trauma and treatment awareness [7]. Even though research in trauma-focused treatment was not initially prioritized in refugee populations [8], trauma-focused treatments have been adapted and proved beneficial for use with refugees with PTSD in the last two decades [9,10,11]. 

Despite a positive development, refugee mental health is still characterized by methodological and clinical underdevelopment, and there are significant gaps in evidence-based mental health interventions for refugees [10,12,13]. This has substantial consequences for affected individuals and their families, with the risk of further marginalization, inadequate access to effective healthcare, as well as being overlooked in the funding of public health services [14]. Meta-analyses on trauma-affected refugees with PTSD include relatively few studies compared to other populations, and these studies often have limited participant numbers and lack control groups for direct comparisons between active interventions [9,11]. Moreover, the literature primarily focuses on PTSD, with less consideration given to comorbid depression, anxiety disorders, and pain [2,13]. Little is known concerning the impact of individual trauma on the family and society and the proportion of the refugee population that, due to treatment access barriers, is left untreated. Long-term, high-quality studies of treatment effectiveness predictors, including complex somatic and psychiatric comorbidity, are needed. As far as diagnostics are concerned, recent research has questioned the adequacy of the PTSD diagnosis for refugees, as many exhibit complex PTSD (CPTSD) symptomatology [15,16,17]. Consequently, there is a need to validate the CPTSD diagnosis in this population and investigate the effectiveness of PTSD interventions for CPTSD.

Factors related to premigration, migration, and particularly post-migration conditions in host countries [12,18] have shown an impact on refugees’ mental health over time [19,20], including a negative impact on the children of trauma-affected refugees [21]. To alleviate mental illness in the refugee population, researchers are thus urged to look beyond health services and examine social determinants of refugee mental health [14,22,23]. 

To potentially motivate the political systems to foster structural changes in developing health services with multi-modal treatment addressing the true complexity of risk and protective factors to effectively reduce health inequities, we need an ambitious research agenda. The research field must establish rigorous systematic data collection standards across individual treatment centers rather than monocenter consecutive outcome studies. In other research domains, there has been a substantial focus on large routine patient-reported outcome measurement systems (PROMs) with standardized questionaries assessing patients’ mental health over the last two decades [24]. Across the world, large-scale PROMs (e.g., in the UK, “Improving Access to Psychological Therapy” [25]; in Australia, “New Access” [26]; in Israel, “Psychiatric Rehabilitation Routine outcome Measurement” [27]; and in Norway, “Prompt mental health care” [28], to name a few) have been disseminated. 

These initiatives aim to monitor clients’ needs and treatment outcomes, informing clinical practice and enhancing service quality while also serving research purposes and ultimately fostering policy changes and rehab paradigm shifts to create services better aligned with users’ needs [24,29]. In Denmark, one other mono-center, routine outcome monitoring system already exists, targeting trauma-affected refugees by collecting data in consecutive pragmatic randomized controlled trials for trauma-affected refugees, Danish Database on Refugees with Trauma (DART) [8]. This initiative demonstrated the feasibility of involving clinical staff in collecting routine monitoring data on trauma-affected refugees for high-quality research purposes. Treatment improvements in trauma-affected refugees in routine care in Denmark and other Nordic countries are, however, quite limited, indicating that enhancing the effectiveness of treatment is challenging when solely focusing on the refinement of traditional treatment components, such as psychotherapy, physiotherapy, and antidepressant medications [30,31,32,33].

On an individual clinical level, PROM data from the Danish Trauma Database for Refugees (DTD) (presented in this paper) can improve communication between clinicians and patients, resulting in improved shared decision making and healthcare [34,35]. At a group level, PROM data from the DTD can be analyzed before and after treatment to provide insights into treatment changes, and in the long term, it can be utilized to guide the development of service delivery and management within the centers. 

On a research level, data from the DTD can test treatment effects in real-world clinical settings with increased sample sizes, improved data quality, enhanced generalizability of results, and the ability to detect rare events or outcomes that may not be observed in single-center studies with smaller sample sizes [36]. Meanwhile, research projects combining DTD with the data from the Danish registers will give unique research possibilities into societal factors and long-term outcomes for the population. In conclusion, national monitoring initiatives across mental health services, such as the DTD, have the potential to promote mental health on a very large scale [37].

The DTD unites six outpatient treatment centers in Denmark for trauma-affected refugees by collecting PROM and sociodemographic data in data waves. It has a two-fold purpose, clinical and research (Figure 1).

The current paper presents the historical background of the DTD, its development process and implementation strategy, and most importantly, it outlines the strategies for the clinical and research utilization of the database. 

### 1.1. The Danish Trauma Database for Refugees (DTD)

#### 1.1.1. Historical Background

The establishment of rehabilitation centers for tortured and trauma-affected refugees marked a significant milestone in recognizing the human rights of refugees and the importance of providing specialized care to survivors of torture and trauma. Danish Institute Against Torture (DIGNITY) was the first such rehabilitation center in Denmark. Following DIGNITY’s establishment, several other national clinics emerged in Denmark in the 1980s [38]. Due to a strong tradition in treating refugee trauma, Denmark currently has a highly specialized treatment infrastructure compared to other Nordic and Western countries. At present, there are eight national rehabilitation/treatment centers for trauma-affected refugees (trauma centers for refugees) and four migrant health clinics in Denmark. Despite the high level of expertise in Denmark, there are still significant treatment barriers for refugees with mental health problems, and the use of mental health services among refugees and ethnic minorities is lower than in the general population [39,40,41,42].

In the Danish healthcare system, the five regions (North, South, Central, Capital, and Zealand) are primarily responsible for the hospitals, general practitioners (GPs), and psychiatric healthcare. Migrant health clinics, which are part of the regions’ general health services, provide treatment for immigrants with complex somatic conditions requiring multidisciplinary, culture-sensitive care. Trauma centers for refugees, on the other hand, are components of the regions’ mental healthcare systems. Yearly, approximately 2500 refugees are offered treatment in these centers. 

A combination of political pressure to document treatment effects, stricter requirements for systematic monitoring, and the potential of developing the field led six of the eight Danish trauma centers to establish the collaboration: DIGNITY, OASIS, Dep. PTSD and Anxiety, the Rehabilitation Center for Refugees (RCF), and the Rehabilitation Center for Trauma Survivors (RCT). Previously, each clinic found it difficult to collect quantitative data rich enough to document treatment effects and to conduct research at the individual center level. Consequently, the leadership of these centers decided to join forces and routinely use the same outcome measurements and establish a joint database. The growth of the DTD collaboration is a continuous focus to optimize the power of the data. Thus in 2022, another trauma center, the Department for Survivors of Trauma and Torture (ATT), joined the collaboration. In the following, we first outline the participating centers and the development of the DTD.

#### 1.1.2. The Trauma Centers for Refugees

All DTD centers use trauma-focused treatments and have a multidisciplinary approach. Of the six centers, three are part of the Danish regional mental health services, while the remaining three are non-governmental organizations (NGOs) operating in agreement with the regions. All staff members are employed; none of the centers rely on volunteer contributions.

The three centers that are part of the regions’ psychiatric healthcare systems are obliged to follow the national guidelines, offering clinical treatment packages for PTSD, typically lasting less than six months. In contrast, the NGOs, which also provide services to families and children, have historically adopted a more multidisciplinary approach with greater flexibility to tailor treatment to patients’ specific needs [43]. For an overview of the treatment centers in DTD, see Table 1.

#### 1.1.3. Patients Included in the DTD Database

While some centers exclusively provide treatment to patients with refugee backgrounds, others also extend their services to immigrants, veterans, or other patient groups with PTSD. Data in the DTD are based on the following inclusion criteria: (1) aged 18 years or older; (2) refugee background; (3) trauma-related mental health problems; (4) permanent or temporally residence permit in Denmark. The exclusion criteria are as follows: (1) acute psychosis; (2) immediate risk of suicide; (3) severe drug abuse. Patients are referred to the trauma centers by other hospital units, GPs, or private psychiatrists for assessment and treatment of traumatic distress. Patients are offered treatment if they meet the inclusion criteria at the specific clinic. Those who do not meet the criteria are subsequently referred to other relevant services in primary care, coordinated with the GP in primary care, or to other treatment in the center if the center treats other patient groups. Only refugees already granted asylum are eligible for treatment in the centers. Refugees without asylum have access to non-specialized treatment in Red Cross Centers.

#### 1.1.4. Implementation Strategy

To successfully implement a multicenter database across national centers, it was essential to establish comprehensive guidelines outlining infrastructure, which include governance policies and procedures, defined leadership responsibilities, resource allocation, data management, clinical training, and a strategy for staff engagement. Having an implementation lead committed to making the collaboration work, investing resources in data management systems and supporting staff, and taking a collaborative approach to designing the PROMS process is pivotal [27,36,37]. Similar to the pre-existing DART initiative, our strategy involved engaging clinical staff in data-collection of research data. We thereby aimed to facilitate ongoing collaboration between researchers, clinicians, and management throughout the development and implementation of the DTD. 

#### 1.1.5. Infrastructure: The Organization and Continuous Development of DTD

Developing robust governance policies and procedures to guarantee the leadership of the DTD and protect data integrity and suitable access took years. Since 2016, the DTD has been governed by a steering committee comprising clinical directors from participating treatment centers, formalized by the first contract in 2018. There is a task group including clinicians, researchers, and data administrators with expertise in clinical assessment and monitoring and an expert panel consisting of external specialists in the field. The external specialists are national clinicians and researchers with expertise in mental health, psychotraumatology, refugee mental health, database research, clinical research, Danish population-based registers, statistics, and epidemiology. Each entity of this structure contributes to the selection of monitoring components and the development of the dataset, which are crucial for maintaining the quality and relevance of clinical research data [44]. New monitoring tools are only included after having been piloted, quality-assured, and approved by the steering committee. Additionally, all research projects and modifications of the DTD setup need to be approved by the steering committee. 

Continuous improvement and quality assurance are ensured through regular meetings, including five yearly DTD task group gatherings and three yearly steering committee meetings in which protocols for collecting data and plans for data management, clinical training, and use of DTD data are discussed and decided. Aggregated data are stored at one clinic, DIGNITY, analyzed in annual reports, and used to inform management and advance treatment development. The task group continuously collects feedback from clinicians to guide the creation of a user-friendly data access system for clinical use in the treatment of individual patients. The six participating clinics jointly finance the maintenance of the database, supplemented by project and research funding from DIGNITY. Future research activities undertaken by DTD will be financed through a variety of research grants, with two such grants already secured. 

#### 1.1.6. Clinical Training and Use of DTD

Training clinicians entailed instructing staff on how to establish a safe, empathetic, and culturally sensitive environment. This involves using plain language and allowing time for questions when obtaining consent to participate, minimizing the risk of re-traumatization. Trained interpreters are used to overcome language barriers, ensuring that participants are well-informed about their rights and the option to withdraw without consequences. Continuous inter-rater sessions and training workshops across centers are utilized to assist clinicians in comprehending and effectively using the DTD data within their clinical practice. 

#### 1.1.7. Patient Confidentially and Ethical Considerations

The DTD prioritizes safeguarding participants’ privacy and ensuring that personal information is securely and confidentially managed in accordance with the General Data Protection Regulation (GDPR). Data security is vital for building trust and encouraging trauma-affected refugees, who often exhibit heightened vulnerability and mistrust, to share sensitive information. This aspect has been a central focus in the implementation strategy from the beginning. 

Data security and integrity encompass policies and procedures for ethical approval, data access, transfer, and storage. The development of the two official contracts (for DW1 and DW2) each required two full years to settle, primarily due to the complex and arduous legal procedures necessary to ensure data security across the Danish regions. Seventeen lawyers and GPDR experts collaborated, developing a solid legal structure for the database and settling the final contract and patient consent information. Clinics maintain ownership and authority over their own clinical data. Patient data from consenting individuals is integrated into the DTD annually. Data are stored in a manner allowing for merging with register data. 

## 2. Methodology 

A crucial aspect of developing a database involves selecting standardized measures that can ensure data quality and comparability across studies. Much effort has been put into the selection of valid and relevant measures that are user-friendly and feasible with real-life clinical practice. All clinics gather information on demographics, socioeconomic status, psychiatric and somatic health, treatment length, as well as trauma exposure, trauma-related head injuries, and post-migratory stressors. Outcome measures include PTSD/CPTSD, anxiety, pain, and disability/general functioning, assessed at pre-treatment, post-treatment, and 9-month follow-up. With the exception of the International Trauma Interview, all instruments utilized are self-report questionnaires. These are completed electronically on an iPad at each center and stored digitally. In the subsequent section, we describe all measurements used in the database. 

Trauma exposure was initially measured using Harvard Trauma Questionnaire, part 1 (HTQ1 Bosnian standard list) [45,46]. HTQ1 is a dichotomous (yes/no) event list of 46 war-related traumatic events. The International Trauma Exposure Measure (ITEM) later replaced HTQ1 to align with the ICD-11 diagnostic criteria [47]. ITEM consists of 21 traumatic life events, where the respondents indicate whether the event occurred during childhood, adolescence, and/or adulthood.

Trauma-related head injury is assessed with Harvard Trauma Questionnaire, part 3 (HTQ3) [45]. The questionnaire contains six dichotomous questions screening for head injuries, oxygen deprivation, and near-death starvation. 

Post-traumatic stress disorder (PTSD) is initially assessed using Harvard Trauma Questionnaire, part 4 (HTQ4). HTQ4 consists of two subscales. A 16-item DSM-IV PTSD symptom-based subscale and a 14-item scale for culturally specific refugee trauma symptoms. The Bosnian version was used as the standard. The HTQ (part 4) has been found to possess high levels of internal reliability (Cronbach’s α = 0.90) and construct and criterion validity. The internal reliability coefficient for HTQ in the present sample was Cronbach’s α = 0.76 [5]. This is later replaced by the International Trauma Questionnaire (ITQ) along with the International Trauma Interview (ITI) following ICD-11 diagnostic criteria for PTSD and CPTSD. Importantly, the ITI is not a screening tool but a clinician-administered diagnostic interview for PTSD and CPTSD [48]. The ITI is carried out by clinical psychologists at each clinic. CPTSD is characterized by Disturbances in Self Organization (DSO) in three clusters: affective dysregulation, negative self-concept, and disturbances in relationships. Consequently, both the ITI and the ITQ have a PTSD part with six core PTSD items and six CPTSD DSO items. Both include six functional impairment items, three related to PTSD and three related to CPTSD. Items are rated on a five-point Likert scale, and patients must meet both PTSD criteria and CPTSD criteria to be diagnosed with CPTSD on the ITI or have probable CPTSD on the ITQ [49]. The ITQ is a twelve-item self-report instrument reporting time since trauma exposure and is a well-validated measure of PTSD (Cronbach’s α = 0.87) and CPTSD (Cronbach’s α = 0.90) [49].

Disability and function are measured using the WHO Disability Assessment Schedule (WHO-DAS II) 12 items [50]. The impact of physical and psychiatric difficulties upon six domains of functioning are assessed: understanding and communicating; mobility; self-care; getting along with others; life activities; and participation in society. The WHODAS 2.0 has been found to possess high levels of internal consistency (Cronbach’s α = 0.86), to be valid for use in different cultures, and for people with physical and mental health problems. The internal reliability coefficient for the WHODAS 2.0 (total score) was Cronbach’s α = 0.97 [50].

Depression and anxiety are assessed with a self-report questionnaire, the 25-item Hopkins Symptom Checklist (HSCL-25) [51], or (HSCL-10) [52], the 10-item version. It has been validated in multiple languages and cultural contexts, making it a useful tool in cross-cultural contexts [43]. The HSCL-25 is valid for use with traumatized refugees. The internal reliability coefficient for the HSCL-25 is high for both depression and anxiety (Cronbach’s α = 0.87 and 0.88) [50]. HSCL-10 has likewise been tested reliable and valid across many samples and cultures [53,54].

Pain severity and pain interference are assessed using the 9-item, short version of the Brief Pain Inventory (BPI) [55]. Patients indicated the location of pains on a human body drawing followed by four items comprising the Pain Severity Subscale: two items on medication use and the degree of medical-induced relief. Lastly, the Pain Interference Subscale measures the negative impact of pain on seven areas (general activity, mood, mobility, work, relationships, sleep, and enjoyment of life). The BPI is widely used in medical and psychiatric populations and has been found to be a valid cross-cultural measure of pain [55,56]. The BPI has been found to possess high levels of internal consistency for both pain severity and pain interference (Cronbach’s α = 0.85 and 0.88) and to be valid for use in medical and psychiatric populations across cultures. The internal reliability coefficients for the pain severity and pain interference scales in the current sample were Cronbach’s α = 0.91 and 0.93 [57].

Recovery is assessed using Brief INSPIRE-O [58]. There is a lack of knowledge on personal recovery and how to promote it among trauma-affected refugees [33]. The 5-item Brief INSPIRE-O has demonstrated adequate psychometric properties [58] and has also been recommended for research and clinical use in a Danish setting [59]. 

Post-migration stressors were initially measured at the 9-month follow-up using the self-developed DIGNITY Life Stress Inventory (LSI). The event list includes 18 events that may have occurred during the last 9 months after treatment and their impact on wellbeing. Later LSI was supplanted by the Post-Migration Living Difficulties Checklist (PMLD) [60]. A short nine-item variation is used covering areas such as access to somatic and psychiatric healthcare, communication difficulties, experienced discrimination, financial difficulties, separation, isolation, conflict with social services, and ongoing unrest in the country of origin.

## 3. Results

### 3.1. The Four Data Waves

#### 3.1.1. Data Wave 0 (DW0)

The early collaboration between national trauma centers was sporadic and restricted. Regular meetings commenced in 2006, as the NGOs (RCT, DIGNITY, and OASIS) were financed by the regions as part of a large structural reform of the Danish welfare system. With all the centers being part of the five regions, collaboration was more feasible. 

The centers aspired to research collaboration; however, several early initiatives failed due to the lack of commitment from the leadership and poor implementation of data collection methods among clinicians. In 2012, DIGNITY established a research-based clinical monitoring battery as part of a Ph.D. research program [61], and four other centers adopted the battery (RCF, OASIS, Dept. of PTSD and Anxiety, and RCT). 

It included validated measures for PTSD, anxiety, depression, pain, and social functioning, along with trauma event lists and sociodemographic variables; thus, the measurements used in DW0 were socio-demographics, BPI, HTQ1, HTQ3, HTQ4, HSCL-25, WHO-DAS II, and LSI. 

The data from 2009 to 2018 was eventually pooled into a database. This was performed retrospectively. Data collecting and cleaning proved time-consuming at every level, even more so merging each center’s individual data sets manually with multiple missing values and inconsistencies, highlighting the need for a shared digital database. 

As can be seen from Table 2, data from 2399 patients were collected in DW0.

#### 3.1.2. Data Wave 1 (DW1)

In 2018, the centers made a formal collaboration agreement, and the establishment of the steering committee and the task group was formalized. A standardized data collection protocol was developed, and a standardized digital format for data collecting was introduced as part of the routine clinical monitoring. This was performed to ensure consistency and comparability of data across the six centers. Workshops were held to gather input on the clinical relevance and feasibility of the measures, and more relevant sociodemographic variables were introduced. 

The measurements used in DW1 were socio-demographics, BPI, HTQ1, HTQ3, HTQ4, HSCL-25, WHO-DAS II, and LSI. 

By the end of DW1, all data was collected digitally, which proved extremely efficient. However, data collecting inconsistencies in the clinical day-to-day life still hampered the process and lowered the data quality, with many incompletes. At the same time, there was a need for a more user-friendly data collection system that could provide clear information about the treatment outcome. Furthermore, new GDPR laws on collecting and storing patient data and the WHO introduction of ICD-11 called for a shift to a second data wave.

During DW1, one clinic, RCT, utilized both HTQ4 and ITQ enabling a future comparison between ICD-10 and ICD-11 PTSD on a subsample of 590 patients with pre-, post-, and follow-up measures. This subsample was to validate construct validity, psychometric properties, and cross-cultural validity when utilizing trained interpreters [62]. Results and feedback from RCT helped recalibrate the DTD prior to DW2. As can be seen from Table 2, data from 2381 patients were collected in DW1.

#### 3.1.3. Data Wave 2 (DW2)

In 2022, a new contract was initiated for DW2, and ATT joined the collaboration. Dept. of PTSD and Anxiety withdrew from the collaboration due to structural changes but continues a close collaboration with DTD with the current DTD research projects. DW2 was developed over a two-year period in close collaboration with clinicians and researchers across the national centers, drawing on continuous feedback and evaluations and lessons learned from DW1. The national expert panel contributed valuable input during the development phase. DW1 and DW2 share multiple instruments and background variables but with some new and updated measurements.

The measurements used in DW2 are socio-demographics, BPI, HTQ3, HSCL-10 (reduced from HSCL-25 to the 10-tem version due to ease of usability), and WHO-DAS II. The new measurements introduced were ITQ, ITI and ITEM, Brief Inspire-O, and PMLD. DW2 runs from 2023 to 2027.

#### 3.1.4. Data Wave 3 (DW3)

DW3 development will begin in 2025. Drawing on insights from DW2, strategic goals will be developed. It is expected to be ready for testing in 2026 and fully implemented in 2027.

## 4. Implications for Clinical Practice and Research

Currently, a series of research initiatives have been commenced with a foundation in DTD. Data from the centers are stored in a manner that allows for research projects to be combined with the Danish national population registers. The registers are a rich source of information on all individuals residing in Denmark, including information on somatic and mental health diagnoses and services consumed, along with other administrative information, such as education and employment. By specific research projects, we are merging the DTD and the register data, which enables more complex analyses focusing on factors within and beyond health services. It allows for the evaluation of the impact of treatment on patients in terms of healthcare use, long-term morbidity, and mortality, as well as education and employment. This research can examine differences between trauma-affected refugees reaching treatment and those not reaching treatment on a range of demographic variables that can further enlighten barriers to treatment. 

We have already published a study of this unique combination of population register and clinically sourced data to document the societal and transgenerational aspects of trauma in refugee populations [63]. Other research projects initiated will be looking at pre- and post-diagnostics, psychopharmacological treatments, and long-term psychiatric, somatic, and socioeconomic outcomes.

A primary objective has been to adjust DTD in alignment with ICD-11 modifications, consequently incorporating the CPTSD diagnosis. A current research project has been launched to create clinical best practice guidelines for PTSD and CPTSD diagnoses with trauma-affected refugees in cross-cultural psychiatry settings. This involves (1) validating the CPTSD diagnosis in a large representative refugee patient population; (2) developing culturally sensitive clinical measures for the use of CPTSD; (3) identifying robust associations between refugee patient characteristics, complex PTSD, and treatment outcomes; and (4) developing manuals and guidelines in collaboration with WHO for assessing complex PTSD reliably in cross-cultural settings with trauma-affected refugees.

Additionally, a culture-sensitive neuropsychological test battery for assessment of cognitive impairment in trauma-affected refugees is currently undergoing validation. This process involves the collection and comparison of neuropsychological data in conjunction with DTD data, with the ultimate objective of documenting and expanding the understanding of cognitive impairment experienced among trauma-affected refugees. 

The close collaboration of the DTD centers reduces the risk of projects being unsuccessful due to contextual and organizational changes. The resources used for research are allocated across several organizations in an otherwise hectic clinical everyday life with scarce resources, protecting the data from being person/organization dependent.

## 5. Clinical and Research Enhancement in the Field

Below we will outline the DTD progress so far with an outset in the clinical and research purpose of the DTD (Figure 1).

DTD has led to the development of treatment for the patients and the centers included in the collaboration. The ability to analyze pre- and post-treatment data to inform clinicians about treatment change enables a dialog with the patient about the experience of treatment change. In the centers that have been part of the DTD from the beginning, DTD has succeeded in enabling clinicians to monitor patients’ progress and plan treatment. Better guidelines are being developed to make the tools more user-friendly for clinicians and to ensure quality feedback is given to patients.

DTD governance has successfully fostered communication among clinicians within the centers. The annual reports have already been used to inform management and advance treatment development. We believe high data quality in DW2 will make DTD an effective tool for shaping service delivery and administration in the future. The DTD is easy to use in everyday clinical practice, and the digital system significantly reduces the amount of time spent on data cleaning. The training of staff has led to professional development and motivation of staff, and the number of clinicians engaged in clinical research has increased since the initiation of the DTD. It is a challenge to train staff systematically across centers, and it has been a challenge to ensure that new DTD partners are trained sufficiently. We are still working to build better guidelines, educational materials, and better access to expert advice for clinicians. 

DTD has a high level of data security, which is essential for engaging both patients and clinicians in research. Given that alliances and trust are crucial factors for positive treatment outcomes, clinicians might be reluctant to participate in data collection if they have concerns about the patient’s security. 

Research-wise, DTD allows for research projects to be carried out that can enhance the understanding of diagnosis and treatment outcomes. Smaller-scale studies have already been carried out in the DTD context, and larger studies have been initiated. 

DTD gives access to a large sample and greater representativeness of data since the clinics are situated over several regions serving diverse populations of trauma-affected refugees. Moreover, the design of the DTD allows for the implementation of quality protocols across the centers, which contributes to high-quality data. This wealth and quality of data make it possible to pose both very broad but also more specific research questions on narrower topics (e.g., make analyses on a cultural level within refugee populations, whose frequency is too low on the individual center level). 

The Danish national population registers allow the DTD to contribute with new knowledge about the large refugee population that is not accessing treatment in Denmark. 

The hope is that funding for clinical research in the field will be made more feasible by the DTD. Long-term, it will be possible to duplicate studies in different settings in Denmark, investigating whether found predictors of treatment outcome are stable across time and settings. 

## 6. Challenges and Future Directions

Carlsson, Sonne, and Silove [8] have previously identified the transition of the field of trauma-affected refugees from a pioneering phase to a second phase of science and rigorous evaluation. We believe a new third phase of multicenter cooperation can bring the field even further.

As a multicenter cooperation, DTD can help meet the current limitations in the field by contributing exceptional research opportunities for testing treatment interventions in clinical settings with larger sample sizes and more representative heterogeneity of the population. At the same time, DTD enables complex analyses of risk and protective factors, barriers, access to treatment, and social determinants of trauma.

Treatment interventions that look beyond health services and address the broader conditions of refugee and asylum seekers’ lives are needed [33,64]. If we want to develop health services with multi-modal treatment which address the complexity of risk and protective factors to effectively reduce health inequities, rigorous systematic data collection standards across individual treatment centers, like the DTD, must supplement monocenter consecutive outcome studies. In the long-term, DTD has the potential to contribute new knowledge, including knowledge on how social and post-migration factors may moderate the ability of refugees to recover from pre-migration trauma, that can be used to develop effective treatment interventions in the future.

Moreover, as a data-driven guidance for routine and clinical treatment planning of the individual patient, DTD has the potential to improve shared decision making and treatment outcomes for all the centers involved and to change the mental healthcare for trauma-affected refugees on a national level. 

There are many challenges and risks associated with working with a multicenter database. Although well-designed multicenter databases can be a valuable resource for advancing health research, it is not surprising that relatively few such databases exist, especially in the field of refugee mental health. Despite the many benefits, addressing these challenges requires significant investment. 

Collecting data of high quality from multiple centers with different practices and collaborating between different centers and professional groups is challenging with substantial risks. At the same time, integrating outcome research in real-life clinical practice is a considerable organizational challenge.

Although all five clinics offer treatment according to the best available evidence-based practice, treatment length, treatment paradigm of choice, clinical staff, and other factors vary greatly and must be closely monitored and taken into account when comparing across the entire sample in the DTD.

The DTD has taken a long time and extensive effort to build. There have been legal, technical, structural, and logistical challenges along the way. The quality of the data is not yet optimal. Several steps have been put into place to manage these limitations, and over time many solutions have been found. Solid governance structures, clear communication, transparency, planning, and involvement have been key factors in getting where we are today. We have learned that an inclusive, bottom-up approach is vital to building a sustainable system that can collect the vital knowledge needed and motivate staff in different parts of the country to carry out data collection in their daily clinical work and drive research projects together. Continuous evaluation is necessary to foresee obstacles arising and ensure the sustainability of the DTD.

## 7. Conclusions

In light of the pressing need for more robust research on trauma-affected refugees, multicenter databases like the DTD can serve as a critical step forward. Our goal is to enhance the knowledge about treatment effectiveness and societal implications of trauma for refugees in high-income countries through representative clinical data. Our ambition goes beyond creating a research database with cutting-edge research. We seek to build bridges between mental health professionals inside and outside the DTD cooperation, including low-resource settings, by sharing data, knowledge, expertise, instruments, and tools. With the current influx of refugees, large systematic multicenter databases and research projects can provide evidence that informs practitioners and decision makers about cost-beneficial treatment interventions and policies that can have positive impacts not only on the population but on the host countries at large. Our hope is to inform and inspire others, leading to more database initiatives in the future that can collaborate with DTD and bring the field even further.

## Figures and Tables

**Figure 1 ijerph-20-06611-f001:**
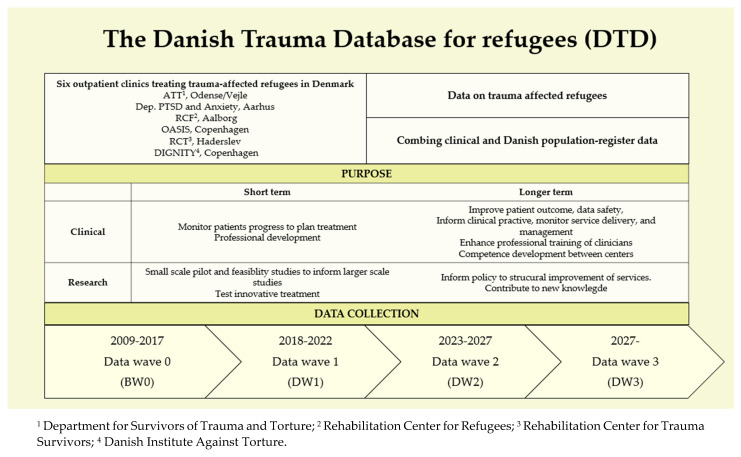
Overview of DTD: centers, purpose, and data collection.

**Table 1 ijerph-20-06611-t001:** Trauma centers included in the Danish Trauma Database for Refugees (DTD).

	NGO (Regions)	Location	Founded	Joined the DTD	Patients	Patients pr. Year (Approx.)	Staff	Treatments
ATT ^1^	Regions (South)	Odense/Vejle	2001/2003	2022	Adult refugees and immigrants with mental health problems and Veterans	550	Psychologists (16), Social workers (8), Physiotherapists (8), Psychiatrists (1–2), Nurses (4)	EMDR, PE NET TFCBT, ACT, MBT
Dep. PTSD and Anxiety	Regions (Central)	Aarhus	1987	2015–2023	Adult refugees and immigrants with mental health problems and Veterans	550	Psychologists (14), Social workers (2), Physiotherapists (7), Medical doctors (2), Nurses (4)	NET, PE, EMDR TFCBT
RCF ^2^	Regions (North)	Aalborg	1992	2012	Refugees with mental health problems	210	Psychologists (6), Social workers (1), Physiotherapists (3), Medical doctors (1), Nurses (1)	NET, TFCBT, PE
OASIS	NGO/in agreement with region (Capital)	Copenhagen	1987	2012	Refugees with mental health problems and their families	200	Psychologists (7), Social workers (4), Psychiatrists (1), Psychomotor therapists (4)	NET, EMDR, ACT, CFT, MBT, SE, TF-CBT, neuroaffective, narrative, and integrative approaches.
RCT ^3^	NGO/in agreement with region (South)	Haderslev	1985	2012	Adult refugees and children with mental health problems	192	Psychologists (7), Social workers (3), Physiotherapists (3), Medical consultants (2)	NET, TF-CBT, PE
DIGNITY ^4^	NGO/in agreement with region (Capital)	Copenhagen	1982	2012	Refugees with mental health problems and their families	160	Psychologists (8), Social workers (3), Physiotherapists (5), Medical doctors (4)	NET, PE, TF-CBT, ACT, CBT

^1^ Department for Survivors of Trauma and Torture; ^2^ Rehabilitation Center for Refugees; ^3^ Rehabilitation Center for Trauma Survivors; ^4^ Danish Institute Against Torture.

**Table 2 ijerph-20-06611-t002:** The DTD collaboration has been collecting data from 2009 and features four distinct data waves. The table below: data waves, data intake, and measurements.

Data Waves	Year	Centers	Measures (Instruments)	N
Data wave 0 (DW0)	2009–2017	DIGNITY RCT Dep. PTSD and Anxiety RCF OASIS	Socio-demographics Pain severity and pain interference (BPI) Trauma exposure (HTQ1) Head injury (HTQ3) PTSD (HTQ4) Depression and anxiety (HSCL-25) Social functioning (WHO-DAS II) Post-migration stressors (LSI)	2399
Data wave 1 (DW1)	2018–2022	DIGNITY RCT Dep. PTSD and Anxiety RCF OASIS	Socio-demographics Pain severity and pain interference (BPI) Trauma exposure (HTQ1) Head injury (HTQ3) PTSD (HTQ4) Depression and anxiety (HSCL-25) Social functioning (WHO-DAS II) Post-migration stressors (LSI)	2381 *
Data wave 2 (DW2)	2023–2027	DIGNITY RCT OASIS RCF ATT	Socio-demographics Pain severity and pain interference (BPI) Trauma exposure (ITEM) Head injury (HTQ3) PTSD (ITI/ITQ) Depression and anxiety (HSCL-10) Social functioning (WHO-DAS II) Post-migration stressors (PMLD) Recovery (Brief Inspire-O)	-
Data wave 3 (DW3)	2028-	-	-	-

* The number of patients in DW1 is a proxy based on the annual number of patients who started treatment from 2018 until 2021. The final number of patients in DW1 is expected during summer 2023.

## Data Availability

Please contact the corresponding author if interested in the data.

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
