# Peer review of "The Danish Trauma Database for Refugees (DTD): A Multicenter Database Collaboration—Overcoming the Challenges and Enhancing Mental Health Treatment and Research for Refugees"

_ijerph, 2023, doi:10.3390/ijerph20166611_

Round 1
Reviewer 1 Report
Review of the Article: "The Danish Trauma Data Base for Refugees (DTD) - a Multicenter Database Collaboration - Overcoming the Challenges and Enhancing Mental Health Treatment and Research for Refugees"
The present study describes an extremely important approach to multicenter collection of health-related data from refugees with the aim of improving the quality of care for this population in various aspects. However, the article has several shortcomings that need to be addressed to provide clearer structure in relation to the research question stated in the title and the key statements of the article.
One crucial aspect that needs improvement is the support for the claim of the high prevalence of physical and psychological complaints among refugees through studies or citations. It is important to substantiate these statements with scientific evidence to ensure the credibility of the article.
The assignment of the "Historical Background" section under "Materials and Methods" does not appear coherent. Since it is not an empirical study, the structure "Introduction – Methods – Results – Discussion" is not appropriate. Clearer structuring should be implemented to provide readers with better guidance.
There is a contradiction between lines 157 and 159 regarding the number of centers participating in the regional Danish healthcare services-DTD collaboration. It should be clarified whether there are a total of five or six centers involved in the collaboration.
The authors should address why only refugees with secure asylum status are treated in the DTD and where refugees with insecure status are treated. It would be interesting to know how the insecure residency status influences the risk of acute psychological decompensation among refugees, as it is known to be a risk factor.
The legal structure and ownership of the DTD are not clearly presented. It should be explained how the funding is ensured and whether the staff are employed and/or volunteers. It would also be helpful to know who the "external specialists" involved in the project are.
The instruments used in the article are presented clearly and comprehensibly. However, it should be clarified whether these instruments are collected decentrally in the participating institutions (e.g., on paper or online) and then transferred to a central server. It should also be indicated who performs the ITI. Additionally, it would be interesting to learn why the LSI was replaced by the PMLD.
In the paragraph about DW0, a reference is made to the instruments of DW1 (line 326). This reference should be clarified to avoid misunderstandings.
Figure 1 is redundant with Figure 2. Figure 2 should be introduced earlier in the text as an overview of the project, and Figure 1 can be removed. The reference to the figure in line 420 should be corrected to establish the correct connection.
It is stated that the DTD started in 2009. It should be clarified whether only one study has been published since then (cf. citation 55) or if further studies based on the collected data have been published. It would also be interesting to know how the collected data are used for monitoring treatment progress in clinical routine.
Furthermore, there are frequent unnecessary spaces throughout the article. The use of citations is inconsistent, with some citations being superscripted and others not. There are missing punctuation marks, such as in line 136, and unnecessary ones in other places, like in line 167 and line 333. Additionally, there are spelling errors, such as "refuges" instead of "refugees" in line 295 and line 550. It is important for the article to undergo editorial revision or proofreading to address these issues. Consistent formatting of citations is also urgently needed.
Abbreviations like DTD in line 100, DIGNITY in line 127, and RCT in line 128 should be introduced before their first use to avoid misunderstandings. It should also be clarified whether DART and DTD are the same and why different abbreviations are used. Furthermore, the abbreviations in Table 2 should be explained to provide better orientation for readers.
Overall, the article contains valuable information about the approach to multicenter collection of health-related data from refugees. However, to improve the quality and clarity of the article, the mentioned shortcomings need to be addressed, and the open questions should be clarified.
The quality of the English language is good. Occasionally words are missing, sentences are incomplete or there are spelling mistakes. This can be corrected by appropriate editing.
Author Response
Dear Reviewer
We would like to express our gratitude for taking the time to review our manuscript titled "The Danish Trauma Data Base for Refugees (DTD) - a Multicenter Database Collaboration - Overcoming the Challenges and Enhancing Mental Health Treatment and Research for Refugees". The feedback is invaluable in strengthening our manuscript and making the message clearer for the readers. We have taken your many excellent comments and feedback into account and made the necessary amendments to the manuscript. Please find our point to point responses to your comments in the attached

Reviewer 2 Report
The authors present a medical database called the Danish Trauma Database for Refugees (DTD), which unites five national centers that provide treatment to refugees affected by trauma. The article describes the background of the database, its development, and implementation strategy, while also outlining the opportunities for use for clinical and research purposes.
My comments are:
The text contained in Table 1 is not appreciated in its entirety.
The text contained in Table 2 is not clearly visible.
The text contained in Table 3 is not clearly visible, also, what does the X mean in column 5?
More DB details could be provided, such as the number of variables and records, general patient statistics, policies, and procedures for the use of the DB by researchers.
Define all acronyms from their first appearance.
It is suggested to include the values of the metrics of the applied instruments.
Author Response
Dear Reviewer
We would like to express our gratitude for taking the time to review our manuscript titled "The Danish Trauma Data Base for Refugees (DTD) - a Multicenter Database Collaboration - Overcoming the Challenges and Enhancing Mental Health Treatment and Research for Refugees". The feedback is invaluable in strengthening our manuscript and making the message clearer for the readers. We have taken your many excellent comments and feedback into account and made the necessary amendments to the manuscript. Please find our point to point responses to your comment in the attached document.

Reviewer 3 Report
Dear authors,
Thank you for the opportunity to review this very interesting narrative article about the DTD. No doubt, the center is doing exceptional work in aiding refugees and treating their mental health issues.
However I am struggling to see the relevance of the submission to IJERPH. For one, what is the study design ? Is this manuscript a narrative review of the DTD or a study protocol ?
In the results section, the authors mention that data have been collected since 2009, however these data are not presented in a scientific manner and not elaborated upon i.e. no data tables/charts etc.
Author Response
Dear Reviewer
We would like to express our gratitude for taking the time to review our manuscript titled "The Danish Trauma Data Base for Refugees (DTD) - a Multicenter Database Collaboration - Overcoming the Challenges and Enhancing Mental Health Treatment and Research for Refugees". The feedback is important in strengthening our manuscript and making the message clearer for the readers. We have taken feedback into account and made the necessary amendments to the manuscript. Please find our responses to your comment in the attached

Round 2
Reviewer 1 Report
I would like to thank you for your manuscript, which I recently had the pleasure of reviewing. You have done an excellent job in making the necessary adjustments that have improved the overall quality of the text. I don't want to nitpick too much, but I would like to point out that there are still unnecessary spaces, inconsistent spelling of quotes, or missing parentheses in some places that should be corrected (including in lines 28, 29, 42, 111, 141, 191, 201, to name examples). Please check the entire text again carefully in this regard. Other than that, everything looks good.
I appreciate your hard work and willingness to improve the manuscript according to my comments. The revisions have helped make the text clearer and easier to understand. I am confident that your manuscript will make a valuable contribution.
Once you have made the mentioned adjustments, I am confident that your manuscript will be ready for further editing. Trusting that the minor changes will be made reliably the text does not need to be seen again by me. Thank you again for your excellent work and cooperation.
Author Response
We are thankful for your kind words and encouragement and appreciate your thoroughness in noting the minor but essential errors related to unnecessary spaces, inconsistent spellings of quotes, and missing parentheses. We apologize for these mistakes and have conducted a thorough review of the entire text to ensure errors have not been overlooked again.
Thank you once again for your attentive review!

Reviewer 3 Report
Dear authors,
Thank you for your clarification and revisions. I understand that this paper has been presented as a methodological paper. However, the article still reads like a narrative and descriptive review of the DTD to me. As such, I will have to suggest rejection of the article.
Author Response
We apologize for any confusion that may have arisen due to the nature and presentation of our paper. We understand that the descriptive approach is not in line with the expected format of a methodological paper. We have, as suggested by Reviewer 1, reclassified the paper as a "Conceptual Paper". We believe that this change in categorization and the new headlines proposed in review round 1, will reduce any remaining confusion and set the right expectations for the readers.
